# Grain Yield Potential of Intermediate Wheatgrass in Western Canada

**Patrick M. LeHeiget [1,2], Emma J. McGeough [2,3], Bill Biligetu [4] and Douglas J. Cattani [1,2,*]**

1. Department of Plant Science, University of Manitoba, Winnipeg, MB R3E 0T5, Canada
2. National Centre for Livestock and the Environment, University of Manitoba, Winnipeg, MB R3E 0T5, Canada; emma.mcgeough@umanitoba.ca
3. Department of Animal Science, University of Manitoba, Winnipeg, MB R3E 0T5, Canada
4. Department of Plant Science, College of Agriculture and Bioresources, University of Saskatchewan, Saskatoon, SK S7N 5C8, Canada; bill.biligetu@usask.ca
* Correspondence: doug.cattani@umanitoba.ca; Tel.: +1-204-474-6071

**Abstract:** Intermediate wheatgrass (*Thinopyrum intermedium*; IWG) is a temperate perennial grass capable of performing in dual-purpose perennial grain cropping systems. It is valued for its ecosystem services and forage yield and quality that can be utilized in many livestock systems. Development for potential perennial grain yield has been the focus of breeding programs for more than two decades, and agronomic management is becoming important, as commercialization of the crop has occurred. This research focused on nitrogen management and intercropping on grain yield and yield components in western Canada. Treatments consisting of a non-fertilized control, an interseeded crop with IWG/*Trifolium hybridum*, and a fertilized treatment (50 kg N ha$^{-1}$) were investigated at four locations. Drought conditions were experienced in some years, resulting in the loss of the interseeded crop at three locations. Fertilization with nitrogen increased grain yield in harvest years two and three and influenced yield components in at least one instance across locations. Third-year grain harvests were higher or equal to year one yield at the two locations harvested, with applied N increasing yield on average by 200 kg ha$^{-1}$ in year three. Inflorescence density is an important yield component after the first production year. The potential for consistent grain yields across three reproductive years was demonstrated, enhancing the potential for sustained productivity.

**Keywords:** intermediate wheatgrass; perennial grain; intercropping; grain production; *Trifolium hybridum*; nitrogen management

## 1. Introduction

Intermediate wheatgrass (*Thinopyrum intermedium*, (Host), Barkworth, and Dewey; IWG) has been gaining interest over the last decade for dual-purpose perennial grain production [1,2]. Perennial dual-use crops allow for the production of a grain crop and the potential to graze the regrowth [3]. They are valued for their flexibility within a production system as well as the potential ecological benefits that they can provide [4]. First cultivated in North America in the early 20th century [2,5], they have been sown on marginal landscapes for erosion prevention, in pastures for grazing, and in fields for hay cropping [6]. Institutions in North America and Europe have been working to increase the grain yield through breeding efforts and management [2]. Advancements have been made, with a cultivar being commercialized in the last few years [1]. However, challenges include increasing the seed yield of this obligate outcrossing species through a combination of genetic selection and better crop management [2]. Grain yield must be increased and be consistent across years to allow for large-scale production and adoption. Improved knowledge of important yield components will aid in this goal.

The ecological benefits of perennials, such as IWG, include the ability to influence soil health, and when intercropped with legumes, the system can employ biological nitrogen

fixation, which provides nutrients and reduces reliance on external inputs [7]. Widely practiced annual systems are experiencing herbicide resistance [8], erosive events [9], and water limitations [10], all of which can potentially be mitigated by perennial cropping with appropriate management [9–11]. At this time, due to the rate of degradation of natural resources, agriculture can benefit from such a land-use change, with some experts suggesting that the implementation and regeneration of perennial systems can slow the rate of organic matter degradation and potentially restore what has been lost [12,13].

Perennial crops are rarely grown compared to their annual counterparts, although the recent success of perennial rice has increased expectations [14]. Grain yield deficit compared to annual grain crops is the primary reason for the lack of adoption [15,16]. As a result, the added economic value of perennial grain crops, such as dual-purpose IWG, for food and feed is critical to their adoption, as ecological benefits are not currently compensated for [17].

With annual cereal production, nitrogen is a limiting nutrient for IWG grain yield [18,19]. A recent report illustrated that the nitrogen required by IWG for grain production does not directly affect the grain but rather the development of the plant increasing the grain yield potential [20]. The addition of synthetic nitrogen in modest amounts has proven to benefit grain yield in IWG stands [18]. Recently, a report demonstrated that 60 kg N ha$^{-1}$ provided a higher grain yield with lower biomass production than a higher rate if no fertility was added, indicating a balance between grain production and growth [21]. Increasing input costs and initiatives to reduce nitrogen losses have led to a demand for alternative sources of nitrogen that can meet crop needs. Research has evaluated the effect of intercropping IWG with perennial legumes in order to benefit from nitrogen sharing in the root zone [19,22–24]. Intercropping has demonstrated some potential to provide more persistent grain yields as the stand ages [22]. As moderate levels of N appear to provide good grain yields [21], the use of legumes appears to have some potential. Therefore, the incorporation of a legume could serve as an effective alternative to limit the need for N inputs throughout the life of the stand.

The developmental pattern for many perennial temperate grasses includes a two-step induction process [25,26]. Reproductive induction of IWG follows that rule; starting with vernalization in the autumn and completing reproductive induction when exposed to the appropriate short nights during the spring months [27]. Factors driving induction in the field cannot be easily controlled; therefore, it is necessary to explore interventions that can influence the activation and recruitment of tillers providing materials for this process. Activation of dormant apical meristems of certain grass species can be affected by the nitrogen level in the soil [28]. This alternative presents potential agronomic avenues to increase the yield potential; however, IWG as a crop has a low reliance on nitrogen to produce aboveground biomass [20], leading to a more complex relationship than initially envisioned.

The Harvest index of IWG can be potentially increased with a better understanding of the development of the crop. *Th. intermedium* is a cool-season, perennial grass that is photoperiod sensitive [29]. The development of rhizomatous structures is a survival strategy that many perennials, IWG included [20], utilized to increase their presence and prominence in a stand [30]. This growth habit can result in the development of reproductive tillers from rhizomes in concert with axillary buds from existing aerial tillers [31], potentially resulting in inflorescences that may contribute to overall grain yield [26]. However, requirements needed for the timing of reproductive induction and the timing of development [31,32] are only now being uncovered [20,27].

The focus of this research is on nitrogen management in the production system and the effect that intercropping with a perennial legume can have on grain yield and yield components in western Canada. Three treatments consisting of a non-fertilized control, an interseeded crop with *Trifolium hybridum,* and a fertilized treatment (50 kg N ha$^{-1}$ post-grain harvest) were evaluated at four locations in Prairie Canada.

## 2. Materials and Methods

### 2.1. Experimental Locations

The experiment was conducted at four locations in western Canada over four growing seasons, and locations, soil types seeding dates, and plot sizes found in Table 1. In 2019, small plot trials were established. The first small plot trial is located at the University of Manitoba's Ian N. Morrison Research Farm near Carman, Manitoba (hereafter known as Carman). The second location is Agriculture and Agri-Food Canada's Brandon Research and Development Centre near Brandon, Manitoba (hereafter known as Brandon). The third small plot location is at the University of Saskatchewan's Livestock and Forage Centre of Excellence located near Clavet, Saskatchewan (hereafter known as Clavet). The large plot grazing study was established at the University of Manitoba's Glenlea Research Station in 2020 near Glenlea, Manitoba (hereafter known as Glenlea), which was grazed by beef cattle in the fall of 2021. However, due to catastrophic flooding in the region in the spring of 2022, which resulted in the loss of the field trial, only one production year occurred.

**Table 1.** Location, soil, and plot information for the four experimental sites in Manitoba and Saskatchewan.

| Location | Location | Soil Factors | | | Seeding Date | Years in Stand | Plot Size |
|---|---|---|---|---|---|---|---|
| | | Series | Texture | Description | | | |
| Brandon * | (49.869264, 99.978605) | Wheatland | Sand | Orthic Black Chernozem | 4 May 2019 | 2019–2022 | 73 m² |
| Carman * | (49.501523, 98.027554) | Denham | Loam | Orthic Black Chernozem | 19 July 2019 | 2019–2023 | 60 m² |
| Clavet ** | (51.935479, 106.381302) | Bradwell | Loam | Dark Brown Chernozem | 15 May 2019 | 2019–2021 | 50 m² |
| Glenlea * | (49.649324, 97.119810) | Scanterbury/Red River | Clay | Gleyed/Gleyed Rego Black Chernozem | 27–29 May 2020 | 2020–2021 | 1.11 ha |

* [33] Manitoba Agriculture, Food, and Rural Initiatives (2010); ** [34] Saskatchewan Soil Information System (2018).

### 2.1.1. Experimental Design

The experiment was arranged in a randomized complete block design (RCBD) with four replicates and was designed to test the effects of the presence of nitrogen in both synthetic and organic forms on grain yield characteristics of dual-purpose IWG systems. It consisted of three treatments: unfertilized intermediate wheatgrass (IWG; *Thinopyrum intermedium*, (Host), Barkworth, and Dewey) monoculture (NOFERT), an IWG monoculture that was fertilized with 50 kg of nitrogen ha$^{-1}$ after grain harvest (FERT), and an IWG-alsike clover intercrop seeded at 50:50 IWG: alsike (INTER). The IWG seed source for these studies is the Syn2 seed of a ten-clone synthetic developed at the University of Manitoba from selections for long-term grain yield capability [35]. At Carman and Brandon, there were two plots of each treatment in each replicate, as a fall and spring biomass and feed quality sample was planned to look at the forage quality of the regrowth. The lack of significant differences between these two regrowth harvest treatments allowed for the removal of this level of treatment.

The trials were conducted under dryland conditions, and each location varied in terms of plot size, as outlined in Table 1, due to limitations with seeding equipment. All sites were soil tested, and pre-planting nutrient levels were brought to the level for perennial forage establishment in western Canada. All plots were planted at 30 cm row spacings for IWG. The intercrop treatment had alsike clover and IWG planted in alternate at 15 cm (IWG every 30 cm). The IWG was seeded at a rate of 6 kg ha$^{-1}$ in all plots, and the alsike clover was seeded at 1 kg ha$^{-1}$ alongside the IWG in the intercropped plots. To reduce the risk of interspecific competition in the emergence and seedling stage, alternate row planting was utilized. The FERT treatments received 50 kg N ha$^{-1}$ of the broadcasted urea post-grain harvest before September in each season, similar to the fertility regime used in the selection nursery for the parentals [35].

### 2.1.2. Experimental Conditions

Drought was a widespread problem throughout the Canadian Prairies from 2019–2022. Some locations experienced it for several consecutive years, while others experienced single growing seasons of drought conditions. Based on the data in Tables 2–5 and Figure 1, each of the locations experienced at least one growing season with below-normal annual precipitation totals, with some receiving slightly over 50% of the 30-year average. It is worth noting that at BRDC in 2020, the precipitation total for the season is near normal; however, over 25% of the total precipitation that season fell in a 24 h period (28 June) and 50% within a seven-day period from 28 June to 4 July 2020 (Figure 1).

**Table 2.** Mean daily temperature and total monthly precipitation for 2019–2022 at Carman, MB, and 30-year average (1989–2018).

| | Mean Daily Temperature (°C) | | | | | Total Precipitation (mm) | | | | |
|---|---|---|---|---|---|---|---|---|---|---|
| Month | 2019 | 2020 | 2021 | 2022 | 30 yr * | 2019 | 2020 | 2021 | 2022 | 30 yr * |
| January | −17.1 | −12.4 | −9.3 | −17.7 | −15.2 | 12.9 | 14.0 | 2.5 | 13.8 | 15.5 |
| February | −20.6 | −12.4 | −17.0 | −18.9 | −12.5 | 35.1 | 1.6 | 3.8 | 20.6 | 13.8 |
| March | −7.0 | −4.9 | 0.0 | −7.6 | −5.3 | 1.4 | 2.6 | 2.6 | 18.9 | 21.4 |
| April | 4.8 | 1.6 | 3.4 | −1.0 | 4.1 | 17.8 | 24.7 | 9.4 | 127.3 | 28.9 |
| May | 9.6 | 10.8 | 10.8 | 10.8 | 11.5 | 36.9 | 27.8 | 27.2 | 110.1 | 78.5 |
| June | 17.3 | 18.3 | 19.3 | 17.5 | 17.1 | 37.9 | 70.2 | 102.9 | 39.4 | 98.0 |
| July | 19.5 | 20.3 | 21.2 | 19.2 | 19.3 | 57.7 | 52.9 | 17.2 | 82.8 | 70.4 |
| August | 18.1 | 18.4 | 18.2 | 19.3 | 18.6 | 61.7 | 24.3 | 78.0 | 48.9 | 68.6 |
| September | 12.6 | 12.3 | 15.7 | 14.3 | 13.5 | 150.7 | 10.8 | 16.5 | 26.7 | 51.4 |
| October | 2.9 | 2.1 | 8.4 | 6.4 | 5.6 | 54.1 | 17.2 | 79.6 | 27.5 | 39.0 |
| November | −5.5 | −3.2 | −1.6 | −4.4 | −4.0 | 16.6 | 4.2 | 23.8 | 14.4 | 23.5 |
| December | −11.4 | −6.9 | −12.7 | −13.9 | −11.7 | 2.7 | 17.5 | 17.6 | 9.8 | 22.3 |
| Mean/Total | 1.9 | 3.7 | 4.7 | 2.0 | 3.4 | 485.5 | 267.8 | 381.1 | 540.2 | 531.3 |

* Observations compiled from Environment and Natural Resources Canada for Carman MB.

**Table 3.** Mean daily temperature and total monthly precipitation for 2019–2022 at Brandon, MB, and the 30-yearr average (1989–2018).

| | Mean Daily Temperate (°C) | | | | | Total Precipitation (mm) | | | | |
|---|---|---|---|---|---|---|---|---|---|---|
| Month | 2019 | 2020 | 2021 | 2022 | 30 yr * | 2019 | 2020 | 2021 | 2022 | 30 yr * |
| January | −18.1 | −13.8 | −10.8 | −18.1 | −16.3 | 20.3 | 11.1 | 8.7 | 36.2 | 17.6 |
| February | −22.6 | −12.6 | −19.4 | −19.5 | −14.0 | 24.4 | 1.8 | 12.1 | 25.1 | 13.8 |
| March | −8.2 | −5.4 | −1.0 | −7.7 | −6.5 | 2.6 | 5.0 | 6.8 | 8.3 | 26.3 |
| April | 4.2 | 0.9 | 3.0 | −0.7 | 3.6 | 14.0 | 23.8 | 13.7 | 44.5 | 25.9 |
| May | 8.9 | 10.4 | 10.0 | 10.2 | 10.9 | 43.2 | 8.2 | 26.2 | 125.8 | 63.9 |
| June | 16.3 | 17.5 | 18.3 | 16.5 | 16.4 | 78.2 | 217.0 | 102.7 | 98.8 | 94.0 |
| July | 19.1 | 20.0 | 20.4 | 19.3 | 18.9 | 41.2 | 59.1 | 29.6 | 85.7 | 66.9 |
| August | 16.8 | 18.3 | 17.1 | 18.9 | 18.0 | 74.9 | 58.6 | 159.9 | 28.4 | 58.6 |
| September | 12.4 | 11.4 | 14.9 | 13.5 | 12.6 | 176.2 | 13.8 | 20.7 | 23.8 | 39.8 |
| October | 2.4 | 0.8 | 7.2 | 5.5 | 4.5 | 50.0 | 10.1 | 23.3 | 16.1 | 32.4 |
| November | 0.6 | −4.5 | −3.2 | −6.0 | −5.3 | 17.6 | 1.5 | 15.2 | 10.3 | 20.0 |
| December | −12.7 | −8.7 | −15.6 | −16.5 | −13.9 | 3.1 | 19.6 | 27.9 | 9.6 | 22.9 |
| Mean/Total | 1.6 | 2.9 | 3.4 | 1.3 | 2.4 | 545.7 | 429.6 | 446.8 | 512.6 | 482.1 |

* Observations compiled from Environment and Natural Resources Canada for Brandon MB.

### 2.2. Measurements

### 2.2.1. Inflorescence Density (INFD)

After anthesis, two (small plots) or eight (grazing location) rows were randomly selected, and 1 m of the row was counted within the plots to determine the inflorescence density. The values were expressed as inflorescences $m^{-2}$.

**Table 4.** Mean daily temperature and total monthly precipitation for 2019–2021 at Clavet, SK, and the 30-year average (1989–2018).

| Month | Mean Daily Temperature (°C) | | | | Total Precipitation (mm) | | | |
|---|---|---|---|---|---|---|---|---|
| | 2019 * | 2020 | 2021 | 30 yr | 2019 * | 2020 | 2021 | 30 yr |
| January | −14.1 | −13.3 | −10.7 | −13.9 | 7.2 | 24.3 | 3.0 | 9.5 |
| February | −24.2 | −10.8 | −19.3 | −15.0 | 11.1 | 3.5 | 2.3 | 6.8 |
| March | −6.1 | −6.6 | −1.2 | −6.2 | 2.7 | 30.2 | 15.2 | 10.8 |
| April | 4.8 | 0.0 | 4.5 | 3.2 | 0.4 | 27.4 | 8.5 | 18.3 |
| May | 9.7 | 11.4 | 10.6 | 11.4 | 4.4 | 47.0 | 41.2 | 39.3 |
| June | 16 | 15.3 | 18.8 | 16.2 | 84.8 | 94.6 | 39.0 | 73.9 |
| July | 17.8 | 18.8 | 21.9 | 18.8 | 67.6 | 34.6 | 8.5 | 51.3 |
| August | 15.4 | 18.6 | 17.8 | 17.6 | 20.3 | 26.5 | 42.2 | 31.2 |
| September | 12.3 | 12.3 | 13.8 | 12.9 | 39.5 | 17.1 | 23.8 | 24.3 |
| October | 0.8 | 1.4 | 5.5 | 3.9 | 11.2 | 7.7 | 6.7 | 16.8 |
| November | −5.5 | −5.7 | −3.6 | −6.6 | 13.1 | 58.5 | 10.2 | 15.7 |
| December | −12 | −9.5 | −18 | −13.3 | 4.1 | 7.3 | 9.9 | 6.7 |
| Mean/Total | 1.2 | 2.7 | 6.5 | 2.4 | 266.4 | 378.7 | 183.7 | 304.6 |

* Observations recorded at the Saskatoon water treatment plant.

**Table 5.** Mean daily temperature and total monthly precipitation for 2020–2021 at Glenlea, MB, and the 30-year average (1990–2019).

| Month | Mean Daily Temperature (°C) | | | Total Precipitation (mm) | | |
|---|---|---|---|---|---|---|
| | 2020 * | 2021 * | 30 yr ** | 2020 * | 2021 * | 30 yr ** |
| January | −12.4 | −10.4 | −16.4 | 7.1 | 3.6 | 19.9 |
| February | −13.1 | −17.7 | −13.2 | 0.9 | 2.9 | 13.8 |
| March | −4.8 | −0.1 | −5.8 | 3.9 | 7.3 | 24.5 |
| April | 2.0 | 3.7 | 4.4 | 15.9 | 24.8 | 30.0 |
| May | 11.0 | 11.5 | 11.6 | 11.6 | 18.3 | 56.7 |
| June | 19.2 | 20.2 | 17.0 | 49.4 | 60.6 | 90.0 |
| July | 21.1 | 22.1 | 19.7 | 43.9 | 9.6 | 79.5 |
| August | 19.3 | 19.0 | 18.8 | 88.7 | 95.1 | 77.0 |
| September | 12.3 | 16.2 | 12.7 | 20.8 | 11.8 | 45.8 |
| October | 2.1 | 9.0 | 5.0 | 11.7 | 52.6 | 37.5 |
| November | −3.3 | −2.6 | −4.9 | 3.4 | 32.7 | 25.0 |
| December | −8.2 | −13.3 | −13.2 | 16.7 | 14.2 | 21.5 |
| Mean/Total | 3.75 | 4.8 | 3.0 | 274.0 | 333.5 | 521.2 |

* Observations recorded at the Government of Manitoba St. Adolphe weather station. ** The 30-year average is based on Environment Canada recordings at the Richardson International Airport in Winnipeg, Manitoba, from 1990–2019.

### 2.2.2. Biomass at Stand Maturity (BIOM)

Immediately prior to harvest or swathing, two randomly selected 1 m rows of IWG were harvested in the small plots, and ten randomly selected 1 m rows of IWG in the large plots were harvested at 4 cm to determine plant biomass at stand maturity. Harvest samples were then bagged and placed on drying beds for 3–5 d to remove any excess moisture. Once the samples were dry, they were weighed to determine the dry matter per unit area.

### 2.2.3. Grain Yield (GYLD)

In the small plot trials, grain yield was determined by total plot yield excluding the outermost rows to serve as treatment buffers. Before the end of August, the year of the harvest, the plots were straight cut using a Wintersteiger plot combine (Wintersteiger Seed-mech, Austria). Samples were then processed by passing through a Westrup Laboratory Air-Screen Cleaner (LA-LS)(Westrup, Slagelse, Denmark) to clean the samples to seed alone and were subsequently weighed to determine the mass of grain harvested per unit area.

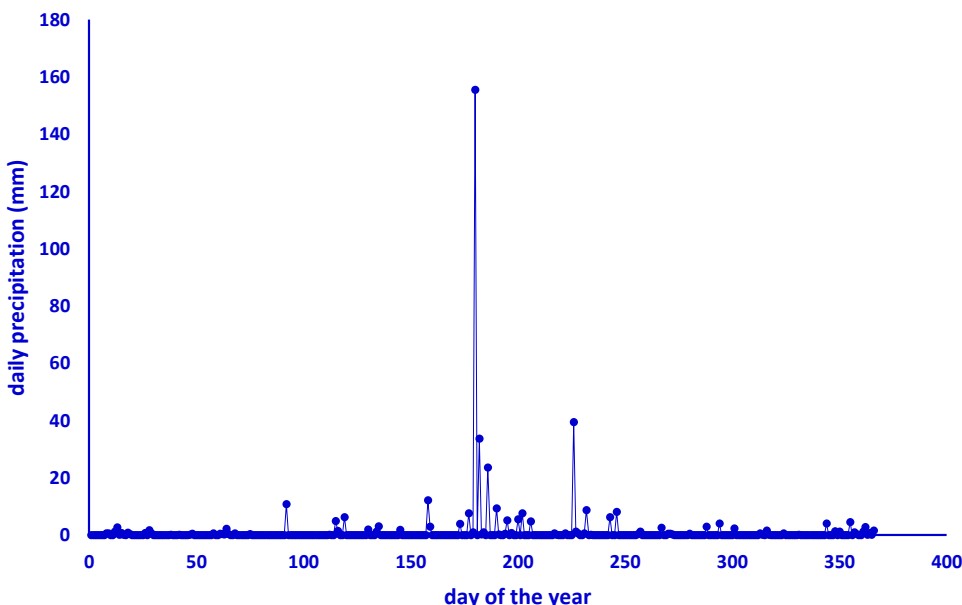

**Figure 1.** Daily precipitation amounts at the Brandon Research and Development Centre in 2020.

In the large plot trial, grain yield was determined by subsampling the large plots. The plots were swathed for harvest to ensure a more accurate yield representation. Then, using a plot combined with a pick-up header, three 10 m lengths of 8 m wide swaths were harvested in each plot. The subsamples were kept separate and put through a Blount Ferrell–Ross-made Clipper M2BC (Blount/Ferrell-Ross, Bluffton, IN, USA) until they were seed only. Subsamples were then combined and weighed to determine the mass of grain harvested per unit area.

### 2.2.4. Thousand Seed Weight (TSW) and Protein (GPROT)

For all trial locations, an approximately 20 g subsample of grain was dehulled by hand by threshing with rub-bars, with hand sieves used to separate seed and chaff and fine material removed with a model 67 Hoffman Seed Blower (Hoffman Mfg, Albany, OR, USA). Once reduced to bare seed, samples were passed through a seed counter to determine the number of seeds in the subsample. They were then weighed to determine the thousand seed weight. Seeds were then milled to pass through a 1 mm sieve on a Wiley Mill (Thomas Scientific, Swedesboro, NJ, USA). Protein was determined by measuring nitrogen in the seed using an FP-528 LECO (Leco Corp., St. Joseph MI, USA) and converted to percent protein using a multiplication factor of 5.75.

### 2.2.5. Harvest Index (HI)

The Harvest index was calculated using the grain yield and the biomass at stand maturity. To perform that, the mass of grain per unit area was divided by the mass of plant material per unit area to determine the proportion of plant material that was grain.

### 2.2.6. Path Analysis

Path analysis was run to look at the influence of components of grain yield on grain yield realized. GYLD, BIOM, and INFD were expressed on a g m$^2$ basis for this analysis. The analysis investigated the direct effects of BIOM, INFD, and TSW on grain yield, and the direct effect of INFD on TSW will be described in the next section.

### 2.3. *Statistical Analysis*

The data were analyzed using SAS Studio software. The first assumption was tested using the PROC GLIMMIX function to generate a dataset of the residuals, and then a univariate procedure was run to test for normal distribution. It was determined that the

residuals had a normal distribution if the Shapiro–Wilk value exceeded 0.90 and the skewness measurement was ±1. Once it was confirmed that the data were normally distributed, the raw data were analyzed using PROC GLIMMIX to verify whether the variances were homogeneous. When the assumptions were met, the analysis was run using the PROC GLIMMIX function to generate an ANOVA. In the cases where normal distribution was not achieved, a lognormal transformation was used if it satisfied the Shapiro–Wilk criterion. If the variances were heterogeneous and the lognormal transformation did not correct it, then a "random _residual_" function was employed to achieve a homogeneity of variance based on AIC and chi-square values (between 0.5 and 2.0). When running the GLIMMIX procedure, the treatments and locations were analyzed as fixed effects, and the replicate factor was nested within the location and considered a random effect. The degrees of freedom were based on the Kenward Rodgers framework, and the Tukey–Kramer LSD test was used to determine statistical significance ($p = 0.05$) between sample means.

Path coefficient analysis was conducted using PROC CALIS in SAS 9.4, and each site year was calculated separately utilizing covariance structure analysis: the maximum likelihood estimation. The model was the direct effects of INFD, BIOM, and TSW on GYLD, and the indirect effect of INFD on TSW was used to look at the indirect effect of TSW on GYLD through INFD.

## 3. Results and Discussion

### 3.1. Grain Yield (GYLD)

#### 3.1.1. Location Effects

Grain yield performance at the individual locations varied throughout production years (Figure 2A) and was impacted by drought and the timing of the precipitation received. A significant location × treatment interaction ($p = 0.017$) was found for the second production year (Figure 3). The first production year showed little yield differences between most locations, regardless of the year of establishment. However, significant differences were recorded for first-year GYLD at Brandon and both Carman and Glenlea, with Brandon yielding half of Carman. In the second production year, there were significant location × treatment interactions, and data are presented by location in Figure 3. In the third production year, there were no significant differences in GYLD between the remaining Carman and Brandon locations. Carman provided mean grain harvests of approximately 643, 845, and 760 kg ha$^{-1}$, with the treatment (FERT) providing the greatest grain yield, which was approximately 200 kg ha$^{-1}$ higher than the other treatments in both the last two years of the study. This relatively consistent performance under conditions of moisture scarcity provides a good base on which to build an agronomic program for the production region. However, annual precipitation amounts were not entirely informative, as was illustrated in Brandon in year 2 (Figure 1). A better understanding of the critical periods of plant development and the influence of the availability of moisture is required to provide for the development of both agronomic practices and decisions surrounding the use of inputs such as fertility. The fertility regime used in the current study was based on the rate used in the selection nursery [35] and other perennial grass seed experiences in Manitoba [36,37].

A cutworm infestation at Brandon led to reduced GYLD in the first production year. Due to institutional COVID-19 restrictions governing entry to the field site, the damage was noticed too late for a successful intervention, and GYLD was impactedThe death of early developing tillers resulted in an impact on grain yield components determining factors at Brandon. Cutworm thresholds for IWG do not exist; however, annual cereals can handle populations of approximately 600 cutworms m$^{-2}$ and 21 cutworms m$^{-2}$ for alfalfa, and under ideal precipitation, threshold values may increase [38]. Ideal precipitation was not the case at Brandon for the 2020 growing season (Table 3). The reproductive induction cycle of IWG may have compounded the impact [27], with new tiller production to replace those lost [39], competing with the reproductive effort. New tillers would likely result in reproductive tillers in the subsequent growing season [8,27,40].

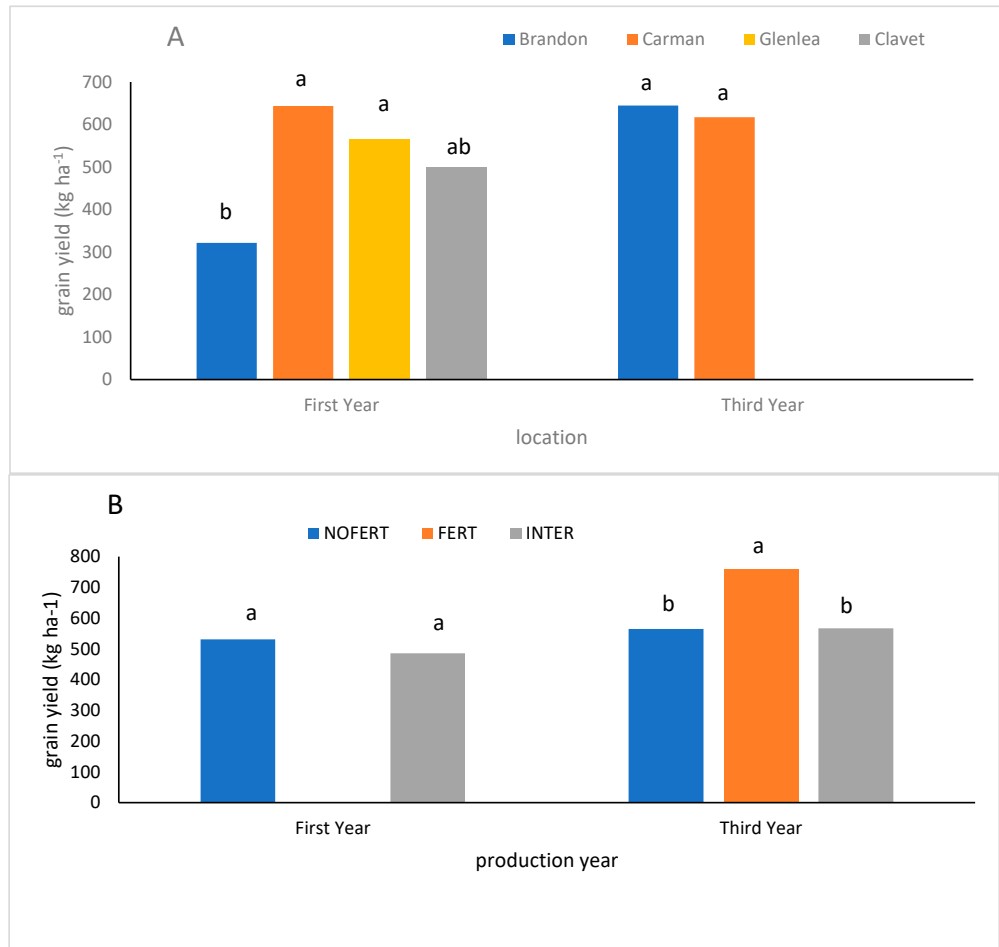

**Figure 2.** Grain yield (GYLD) of intermediate wheatgrass (IWG) for the four locations (**A**) and three treatments+ (**B**) in western Canada for the first and third reproductive years. The same letter over bars within individual locations is not significant using Tukey–Kramer LSD at $p$ = 0.05.

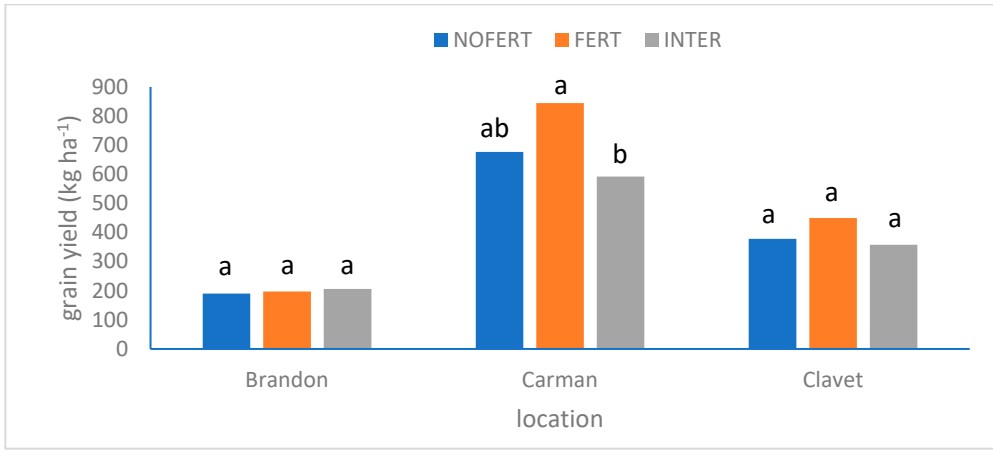

**Figure 3.** Location × treatment effects for grain yield in the second production year. The same letter over bars within individual locations is not significant using Tukey–Kramer LSD at $p$ = 0.05.

In the following reproductive year, the moisture conditions were less favorable at Brandon (Table 3), and GYLD remained much lower than at other locations. The potential thinning experienced in 2020 may have inhibited adequate tiller recruitment to produce a satisfactory GYLD the following year.

By the third production year, the GYLD at Brandon did not differ from Carman, with the latter remaining relatively consistent across all years. It is possible that there was little cutworm activity resulting from an increase in beneficial insect pressure [38,41]. The potential cumulative impact of the above, coupled with the greater precipitation, resulted in yields at Brandon that were equal to Carman.

### 3.1.2. Treatment Effects

GYLD was not significantly influenced by treatment in the first production year (Figure 2B). There were only two treatments in this production year, as FERT was first applied after the first harvest. In the second production year, Carman recorded a FERT GYLD more than four times greater than all Brandon treatment means (Figure 3) and significantly greater than the INTER treatment. Brandon yielded the lowest means for all the treatments, with this location and Clavet presenting no treatment differences in GYLD, while significant GYLD differences were found in treatments at Carman in the second production year. In the third production year, FERT resulted in a significantly higher GYLD than the other treatments (Figure 2B). The FERT GYLD was almost 200 kg ha$^{-1}$ greater than the other treatments, which in turn did not differ. In the third production year, GYLDs were greater at Brandon than the two previous years combined, while remaining relatively static at Carman across all years. NOFERT and INTER were statistically similar in all years, indicating that there was little grain yield advantage to seeding IWG alone.

The differences found at Carman in the second and third production years and at Brandon in the third production year illustrate the potential impact of added fertility. Nutrient sharing between the IWG and alsike clover for the INTER treatment was not sufficient to enhance GYLD. Competition has been an issue in numerous studies when legumes were intercropped in the early stages of the IWG grain production cycle [19,22–24]. Although this issue arises early in stand life, research indicated that intercropping provided a GYLD benefit in the third production year in some instances [22]. By the end of the first production year at Brandon and Clavet, there was little alsike clover remaining in INTER and at Carman by the end of the second production year. The INTER and NOFERT treatments were similar, and no GYLD benefit was observed through intercropping. This lack of INTER effect is contrary to what was expected, with a reported correlation between legume biomass in the first production year and the GYLD of IWG in the third production year [19]. The implication may be that a positive effect requires that the legume survives into the third production year, continuing to contribute nitrogen, thus helping meet the nitrogen demand of the IWG stand.

Grain yield for the third production year demonstrated the impact of FERT and the potential for consistency of production, with FERT yielding approximately 200 kg ha$^{-1}$ more than the other treatments. Other reports on the impact of synthetic nitrogen applications at similar rates have resulted in increased GYLD in later production years [18,42]. Nitrogen rates higher than those used in this study generally did not result in greater GYLD [20]. Nitrogen requirements and the timing within the crop still require greater fine tuning with the growth environment. Nitrogen and its timing in the current study had a positive effect on GYLD and is expected to be a key factor in maintaining grain productivity [20,28]. Rates may differ due to growing season length, moisture amounts and timings, and any harvesting of vegetative biomass for feed or by animals.

Grain yield at Brandon fluctuated while at Carman, a consistent yield was realized through the three production years. Consistent or increasing GYLD in year three, coupled with the impact of fertility applications, indicates a long-term yield potential for this crop. This is the first report we are aware of indicating this consistency, and it may be attributed to the selection criteria of taking materials through three grain harvests prior to final selections being made [35]. As expected, drought appears to have had a large impact on GYLD at locations throughout these studies. Precipitation patterns may be as important as precipitation amounts and may be critical to the maintenance of GYLD potential in IWG. Brandon received approximately 90% of its average 30-year precipitation amounts

(Table 3) in 2020. However, the timing of the precipitation is important, with 217 mm in June of 2020 accounting for >50% of the annual amount. Between 28 June 2020 and 4 July 2020, 217.4 mm of precipitation was recorded, again >50% of the total annual precipitation (Figure 1). Future research, possibly a synthesis from reported studies, may determine critical periods for moisture. In Manitoba, where four to six months of winter can be experienced, the timing of precipitation to coincide with crop requirements, especially during periods of active growth, will likely inform the timing of agronomic applications for crop productivity.

### 3.2. Harvest Biomass (BIOM)

Biomass yields were different between years but did not demonstrate any significant differences between the locations and treatments throughout the study (Figure 4). Biomass yield at all locations decreased in the second production year, with drought conditions experienced at each site. Brandon did not significantly differ in BIOM from the other locations in the first two production years where cutworm damage occurred. This suggested that although insect damage reduced GYLD, maintaining a comparable BIOM was possible for IWG. The plants were likely able to generate new vegetative tillers to compensate for the loss of reproductive tillers due to the cutworms [39,43]. The timing of herbage removal and the presence of adequate moisture affects tillering potential, with early removal in the growing year resulting in higher potential tillering [39].

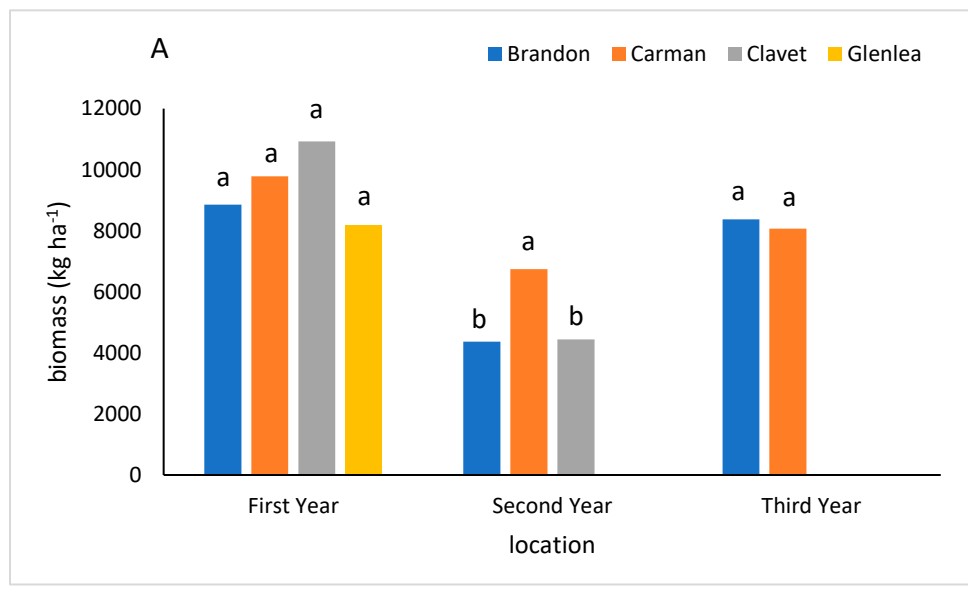

**Figure 4.** Biomass at the harvest of intermediate wheatgrass (IWG) for the four locations in western Canada for the first, second, and third reproductive years. The same letter over bars within individual production years is not significant using Tukey–Kramer LSD at $p$ = 0.05.

Biomass yield in the third production year was at levels approaching the first production year, which was likely due to the better moisture conditions experienced. Treatments were not significantly different, suggesting that the addition of nitrogen the previous fall did not affect BIOM, which is consistent with another report [44]. However, it is worth noting that a positive BIOM response to spring fertilizer additions has been reported [42]. Fall application was used in this study; therefore, the timing of application may be a necessary consideration when looking at the forage production aspect of the dual-purpose system.

### 3.3. Harvest Index (HI)

Unlike the previous two measures upon which the HI is calculated, the HI showed significant differences among locations in the first production year (Figure 5). Glenlea had an HI that was almost twice what was recorded in Brandon; however, the latter location

experienced reproductive tiller predation as previously outlined. Clavet did not differ significantly from Brandon, and Carman did not differ significantly from Glenlea. In the second production year, Carman and Clavet were not statistically different, and while Carman increased year over year, it was significantly less than the other locations. In the third production year, the HI at Brandon was similar to Carman.

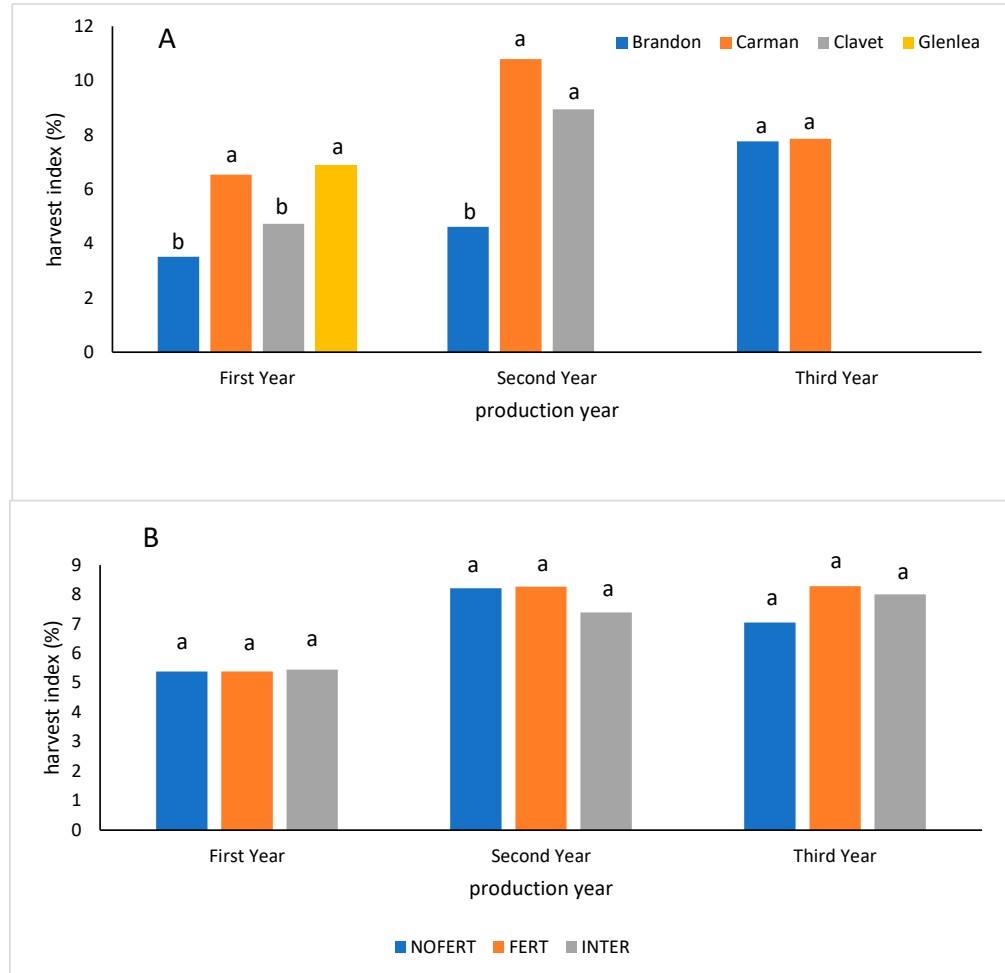

**Figure 5.** Harvest index (HI) of intermediate wheatgrass (IWG) for the four locations (**A**) and three treatments (**B**) in western Canada for the first, second, and third reproductive years. The same letter over bars within individual production years is not significant using Tukey–Kramer LSD at *p* = 0.05.

The Harvest index was likely influenced by drought conditions that were present during the experiment. Additionally, there are a few reports indicating that the harvest index of IWG shows consistent reductions as the stand ages [5,22,45]. This reduction has been attributed to increased BIOM, while GYLD remained stagnant or decreased [45]. Looking at the trends previously outlined, BIOM declined in the second production year and increased in the third at both Brandon and Carman. However, the HI at the latter location increased in the second year and decreased in the third, remaining higher than the first production year. The harvest index at Brandon continuously increased, doubling in the third production year compared to the first production year (Figure 5). This location was impacted in the first year by cutworm damage and drought in the second; however, as conditions became more favorable, the HI responded positively.

Significant differences between treatments were not found in the first and second production years for the HI. In the third production year, the HI for the FERT treatment was significantly greater than NOFERT. The INTER treatment did not differ from the other treatments. Nitrogen fertilizer application in older IWG grain stands was found to increase

the HI at one of five locations in a US study, with two locations experiencing an increase in the HI with an alfalfa intercrop and the other two locations showing little difference between treatments [22]. In spring wheat, the HI increased significantly with a higher rate of nitrogen fertilizer (200 kg ha$^{-1}$; [46]). Therefore, this relationship requires further research in IWG.

### 3.4. Thousand Seed Weight (TSW)

In the first production year, TSW at Carman and Clavet was significantly greater than Brandon and Glenlea (Figure 6A). In the second production year, while TSW decreased at all locations harvested, Carman and Clavet remained significantly higher than Brandon. Both Carman and Clavet experienced notable moisture limitations during that production year (Tables 2 and 4); however, Brandon did not appear to experience a detrimental moisture limitation, based on data presented in Table 3. Notwithstanding, as previously noted, the annual value is misleading. In the third production year, Carman and Brandon produced a similar TSW, slightly lower than the TSW recorded at Carman in the first production year. This increase fits the moisture change experienced at Carman and aligns with other increases experienced at Brandon in the third production year.

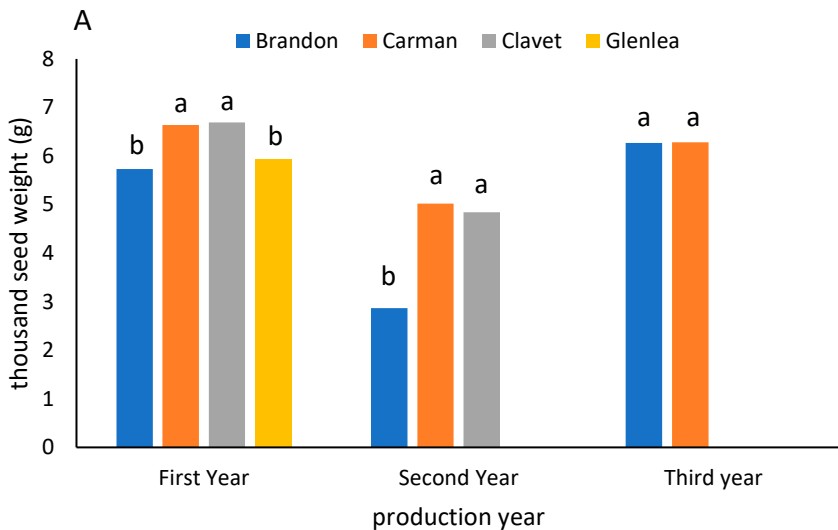

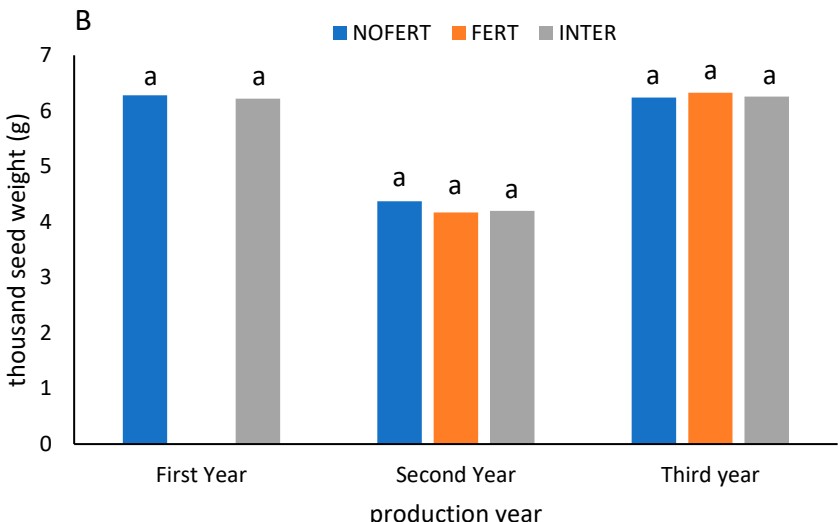

**Figure 6.** Thousand seed weight of intermediate wheatgrass (IWG) for the four locations (**A**) and three treatments (**B**) in western Canada for the first, second, and third reproductive years. The same letter over bars within individual production years is not significant using Tukey–Kramer LSD at *p* = 0.05.

No treatment effects were found through the course of this study for TSW, which appears to indicate that the growth environment is likely the primary influence on the TSW in IWG (Figure 6B). A connection between nitrogen application and seed weight has been reported in both annual grain [47] and perennial ryegrass seed production [48]. Seed weight was previously noted to be related to seed yield in IWG on space plants [49]; however, the current study is from a crop perspective. More research is needed to determine the influences on the TSW in IWG crop systems.

### 3.5. Inflorescence Density (INFD)

Treatment differences for grain yield were matched by similar increases in INFD, which varied significantly between locations in the first production year (Figure 7A). At Glenlea, INFD was significantly higher than the other three locations, and more than double what was observed at Brandon. In the second production year, the INFD increased at all locations, and differences between treatments changed. At Carman, INFD was significantly higher than Brandon, with Clavet not differing from either of the other locations. This is consistent with the Brandon GYLD trends outlined previously and the moisture limitations at Clavet. First-year INFD at Glenlea was in the same calendar year as the second production year at the other locations; therefore, the increase seen in the second production year may be due to environmental conditions and not stand age. In the third production year, treatment had the only significant effect.

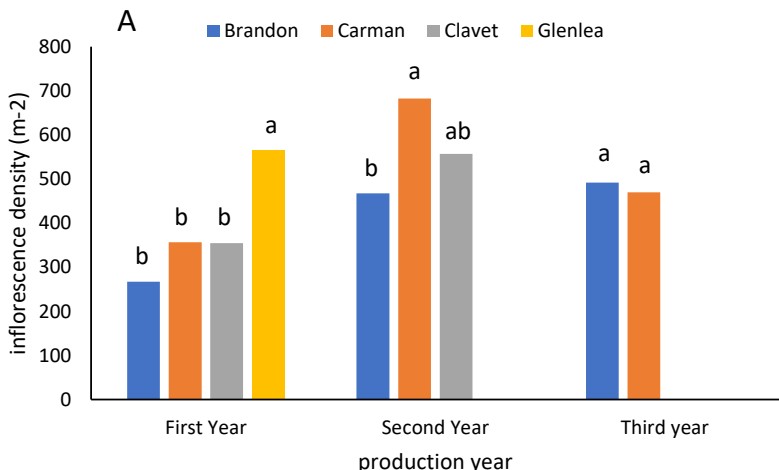

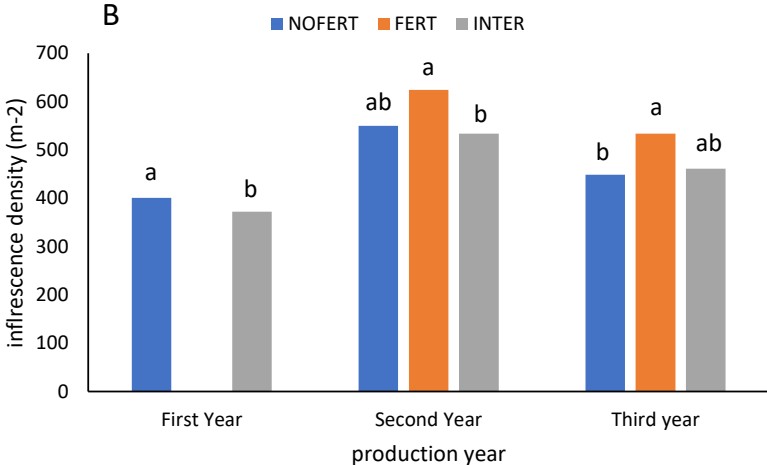

**Figure 7.** Inflorescence density of intermediate wheatgrass (IWG) for the four locations (**A**) and three treatments (**B**) in western Canada for the first, second, and third reproductive years. The same letter over bars within individual production years is not significant using Tukey–Kramer LSD at $p = 0.05$.

The application of fertilizer in the fall appears to increase INFD, increasing the grain yield potential, and in this study, treatment had a significant effect on INFD in all production years, with NOFERT resulting in greater INFD than INTER in the first year of production, though it was likely competition mediated (Figure 7B). In the second production year, FERT was significantly higher for INFD than NOFERT, while INTER did not differ significantly from either treatment; again, competition between species is the likely factor. The treatment differences in these two years support the idea of a drought-induced competition between alsike clover and IWG in INTER treatment, with the amount and timing of fall precipitation potentially being a critical factor. As a species with a vernalization requirement, fall regrowth should provide tillers for conversion to reproductive primordia over the winter. The INFD in the third production year was significantly greater for FERT versus NOFERT and was similar to the increase in GYLD. However, the gap between FERT and NOFERT was greater for GYLD, indicating that fall fertility was influenced more than INFD. Good plant nutrient status appears to enhance seed set, as TSW was similar between treatments, especially where water was not in scarce supply. Further studies, including density manipulations, may provide a better understanding of these relationships. One other possibility could be the loss of the legume that provided some nitrogen, which benefitted the IWG in INTER, providing an advantage over the NOFERT treatment. This postponed benefit is similar to findings previously outlined in the GYLD discussion [19], where earlier legume biomass was correlated to later IWG GYLD, but unlike GYLD, a latent effect was observed. A significant increase in bud activation in $C_3$ perennial grass species has been reported when increasing the nitrogen rates [28]. This concept is further supported, as it has been noted that the number of inflorescences increases with the rate of nitrogen application in grasses that require dual induction, similar to IWG [26,50–52]. The INFD observed at Brandon was similar to Carman in the third production year.

### 3.6. Grain Protein (GPROT)

Differences in GPROT were inconsistent during both growing seasons where protein was measured. In the first production year, significant differences were only found at Glenlea, where the GPROT of INTER was significantly greater than NOFERT (Figure 8A). Successful establishment and regrowth of the alsike clover (Figure 9), unlike the other locations, appears to benefit IWG, similar to successful nitrogen transfer by red clover to IWG [19]. In the second production year, no significant differences were found in treatments at Carman (Figure 8B). At Brandon, FERT was higher than NOFERT and INTER, while at Clavet, FERT was higher only than INTER. In general, the increases observed with FERT imply that the application of nitrogen has a significant positive effect on the GPROT in IWG, similar to winter wheat [53]. Grain protein at Clavet in the second production year was at the high end of the reported values in IWG [54]. It is possible that drought concentrated the grain protein, which is again similar to what was observed in winter wheat [53] and is likely due to reduced seed size.

### 3.7. Path Analysis

A path analysis was carried out on a site-year basis. No consistent effects were noted across all site years (Table 6). In the first production year, biomass m$^{-2}$ had the largest direct effect on grain yield m$^{-2}$ for all sites; however, Clavet did not have any significant direct effects. In the second and third production years, inflorescence density had a significant direct effect on seed yield for all site years. These results are similar to results from the selection nursery upon which the plant materials used in this study were developed [49]. TSW had a significant direct effect on grain yield in two site years only. TSW had a significant

direct effect on inflorescence density in two years, which is an indication of some yield component compensation. Inflorescence density as the stand ages appears to be a good indicator of yield potential within the stand. This has been previously seen in another cool-season perennial grass species (*Agrostis stolonifera* L.) produced in Manitoba [37]. Tiller development after heading in *A. stolonifera* provides for the vegetative tissue that undergoes vernalization in that species [32], effectively dictating yield potential. Additional work is required to validate this in IWG; however, this work would lead to a greater understanding of development and ultimately to better agronomic practices.

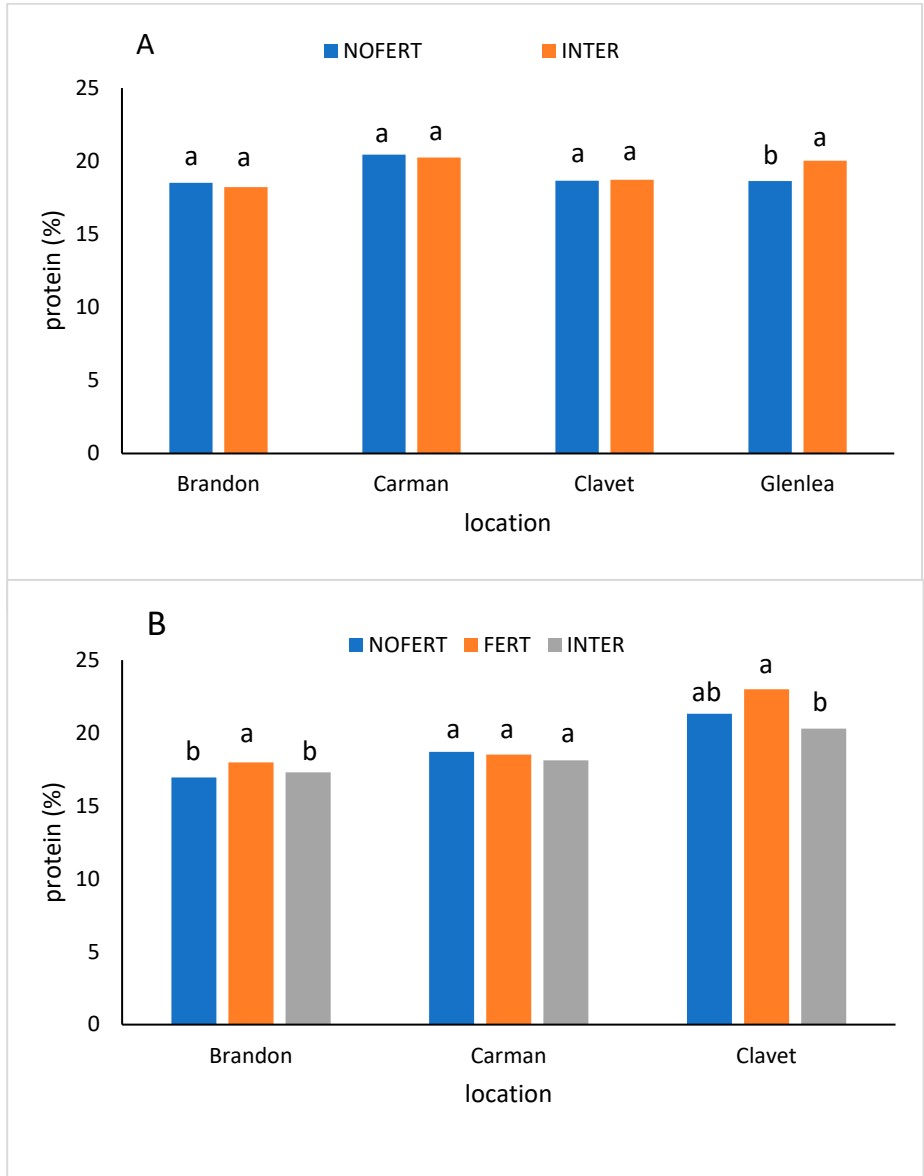

**Figure 8.** Grain protein % for the treatments at the locations in the first production year (**A**) and second production year (**B**) of intermediate wheatgrass (IWG) for the four locations in western Canada. The same letter over bars within individual locations is not significant using Tukey–Kramer LSD at $p = 0.05$.

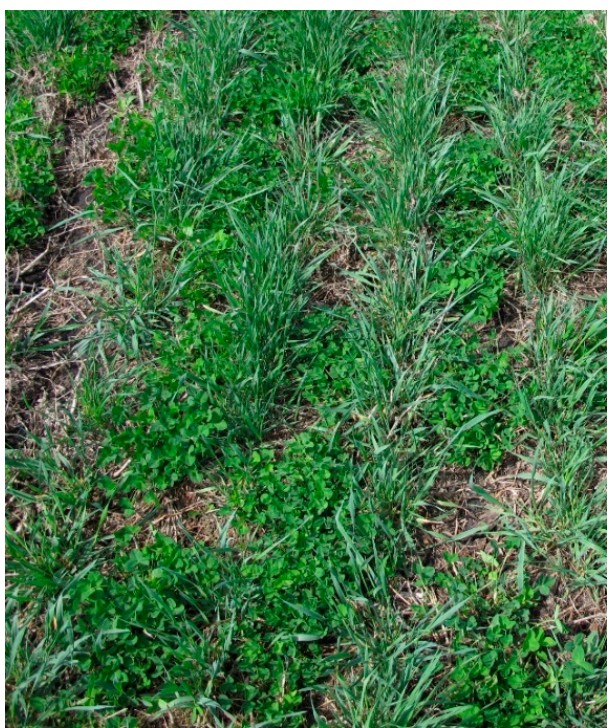

**Figure 9.** The interseeded treatment at Glenlea illustrates the development of intermediate wheatgrass and alsike clover in May of the first production year (2021).

**Table 6.** The direct effects of inflorescence density m$^{-2}$ (density), biomass m$^{-2}$, and thousand seed weight (TSW) on seed yield m$^{-2}$ and thousand seed weight on inflorescence density for each site year studied.

| | Direct Effect on Yield m$^{-2}$ | Direct Effect on Density m$^{-2}$ | Direct Effect on Yield m$^{-2}$ | Direct Effect on Density m$^{-2}$ | Direct Effect on Yield m$^{-2}$ | Direct Effect on Density m$^{-2}$ | Direct Effect on Yield m$^{-2}$ | Direct Effect on Density m$^{-2}$ |
|---|---|---|---|---|---|---|---|---|
| | **Brandon** | | **Carman** | | **Clavet** | | **Glenlea** | |
| | | | | First production year | | | | |
| Density m$^{-2}$ | 0.138 | | 0.173 | | 0.383 | | 0.445 ** | |
| Biomass m$^{-2}$ | 0.766 ** | | 0.491 ** | | 0.253 | | 0.569 *** | |
| TSW | 0.059 | 0.452 ** | 0.343 * | −0.077 | 0.169 | 0.139 | 0.231 | 0.122 |
| | | | | Second production year | | | | |
| Density m$^{-2}$ | 0.292 * | | 0.717 *** | | 0.774 ** | | | |
| Biomass m$^{-2}$ | 0.030 | | 0.179 | | 0.068 | | | |
| TSW | 0.813 *** | −0.215 | −0.048 | 0.142 | −0.117 | 0.315 | | |
| | | | | Third production year | | | | |
| density m$^{-2}$ | 0.312 * | | 0.758 *** | | | | | |
| biomass m$^{-2}$ | 0.582 *** | | 0.051 | | | | | |
| TSW | 0.100 | 0.080 | 0.008 | 0.695 *** | | | | |

*, **, *** indicate significance at $p$ = 0.05, 0.01, and 0.001 levels.

### 3.8. Other Considerations

The western Canadian provinces are the major producers of forage seed in Canada [3]. Providing producers with an agronomic program that aims to provide a sustainable grain yield across years is a good first step to commercializing this new crop. Crop age and soil type did not appear to impede GYLD through three production years, and the growth environment had a major role. However, GYLD is a limitation to the uptake of IWG as a perennial grain [55], and breeding this species is in its infancy [2,56] compared to its annual counterparts. As such, lower yields would need to demand a higher commodity price to

be financially viable for producers. Breeding progress on many traits may be achieved more rapidly than grain yield, as we are just beginning to determine the characteristics that contribute to grain yield sustainability. Additionally, the protein concentrations seen in the current study (approximately 17–23%) add a potential premium for this grain.

Net zero carbon is currently a major aim for agriculture in Canada by 2050, although agriculture as a provisioning activity can justify some carbon usage. Sustainable productivity is a key goal for agricultural systems and allows for the accumulation of more ecosystem services (e.g., carbon sequestration potential). The perennial grain system used in this study with its low inputs could help reduce agriculture's carbon footprint, especially in marginal areas where annual crop production is limited. Carbon sequestration is one of the purported benefits of perennial grain systems [4,16]; however, it will be dependent upon a range of factors including the duration of the stand, soil type, and environmental conditions. Enhancing or at worst maintaining soil health and quality should be seen as positive outcomes of perennial grain production, and this study indicates the potential for consistent reproductive efforts enhancing the outlook for long-term crop stands and the potential benefits derived from them [4]. The quantification of these benefits may provide incentives for producer uptake, especially if monetary values can be attached (e.g., carbon credits). Changes in policy to allow this will likely be slow; however, they also allow for the development of higher-yielding cultivars and better agronomic practices.

Economic return is the primary driver for producer adoption of a new crop. As the yield of IWG is still low compared to annual grains, other incentives are likely needed for widescale adoption. Carbon credits of carbon sequestration, soil health benefits, and reduced tillage have potential economic value to growing this crop; however, they are not recognized (e.g., carbon credits) or available in the two provinces in which we tested this crop. The potential to garner support for and the assignment of an economic value to the individual product or practice is dependent on the value society will place on them. A change in the attitude [57] or the measurement and costs [18] of crop production will be required. This will need societal approval, but it will be critical to have producers on board and propose these changes rather than have them implemented from government directives.

## 4. Conclusions

The range of soils in which these studies took place did not disadvantage the crop, and drought and insect infestation were the major factors limiting grain yield. Nitrogen resulted in increased GLYD, especially as the stand aged, which was in part due to increased inflorescence density; however, the loss of the interseeded alsike clover due to drought resulted in an inability to evaluate the nitrogen transfer potential from a perennial legume.

Biomass influenced grain yield in the first production year, and inflorescence density had the greatest direct effect on grain yield in subsequent production years.

Greater in-season precipitation in the third production year led to GYLD similar to the first production year, indicating a consistent perennial grain yield potential of IWG in western Canada. Further development of our understanding of the crop and the factors that influence yield potential and yield is greatly needed to provide producers with agronomic programs that may provide a sustainable grain yield of this dual-purpose crop and is a good first step to commercialization.

**Author Contributions:** Conceptualization, D.J.C., E.J.M. and B.B.; research, D.J.C., P.M.L., E.J.M. and B.B.; methodology, D.J.C., E.J.M. and B.B.; formal analysis, P.M.L. and D.J.C.; investigation, P.M.L., D.J.C., E.J.M. and B.B.; data curation, D.J.C.; writing—original draft preparation, P.M.L.; writing—review and editing, D.J.C., E.J.M. and B.B.; funding acquisition, E.J.M. and D.J.C. All authors have read and agreed to the published version of the manuscript.

**Funding:** This research received funding from the NSERC Strategic Program, grant number STPGP 521846-18, Manitoba Beef Producers, and the MITACS Accelerate Program. The authors acknowledge the in-kind contribution of Agriculture and Agri-Food Canada and the Brandon Research and Development Centre for land, equipment, and other resources for the small plot research conducted at this site.

**Institutional Review Board Statement:** Not applicable.

**Data Availability Statement:** Data can be made available by contacting doug.cattani@umanitoba.ca.

**Acknowledgments:** The authors would like to thank the following people for their technical support in conducting this research: Ardelle Slama, William Giesbrecht, Mae Elsinger, Rhonda Thiessen, and Dashnyam Byambatseren.

**Conflicts of Interest:** The authors declare no conflict of interest.

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
