# Peer review of "Grain Yield Potential of Intermediate Wheatgrass in Western Canada"

_agriculture, doi:10.3390/agriculture13101924_

Round 1
Reviewer 1 Report
As a reviewer for this journal, I have reviewed the paper titled "Grain yield potential of intermediate wheatgrass in western Canada." The paper explores the grain yield potential of intermediate wheatgrass (Thinopyrum intermedium; IWG) in western Canada, focusing on nitrogen management and intercropping. I appreciate the paper's scope, which addresses an important aspect of agricultural research, particularly in the context of sustainable grain cropping systems. The manuscript provides a concise overview of the research objectives, methods, and key findings. Here are some comments and questions for further consideration:
1. The control group (NOFERT) represents an unfertilized monoculture of IWG. Could the authors explain the rationale for choosing this specific control group and how it relates to the research objectives?
2. The intercrop treatment involves planting IWG and alsike clover in alternating rows. Could the authors provide insight into the reasoning behind this specific planting pattern and how it might affect interactions between the two species?
3. It's mentioned that the FERT treatment received nitrogen fertilization post-grain harvest before September each season. Could the authors elaborate on why this timing was chosen and how it aligns with IWG growth cycles and nutrient uptake patterns?
4. Is there information on the number of replicates for each treatment and how randomization was applied within the randomized complete block design? Understanding replication and randomization is crucial for assessing the robustness of the experimental design.
5. All references should be in [ ] instead of ( ).
6. There are so many incomplete sentences. Please complete the sentences. such as in section 3.1.1, “The potential cumulative impact of the above, coupled with the.....”
7. In some places, authors mentioned kg/ha; in others, they mentioned kg ha-1. Please use the same format throughout the MS.
8. In Conclusion
remove fullstop from 4.1 .Conclusion
The conclusion is too long, even longer than the abstract; it should be shorter and concise.
The phrase "Favorable moisture conditions" is somewhat vague. Consider specifying what exactly "favorable moisture conditions" means or entail for better clarity.

Author Response
Reviewer 1:
As a reviewer for this journal, I have reviewed the paper titled "Grain yield potential of intermediate wheatgrass in western Canada." The paper explores the grain yield potential of intermediate wheatgrass (Thinopyrum intermedium; IWG) in western Canada, focusing on nitrogen management and intercropping. I appreciate the paper's scope, which addresses an important aspect of agricultural research, particularly in the context of sustainable grain cropping systems. The manuscript provides a concise overview of the research objectives, methods, and key findings. Here are some comments and questions for further consideration:
1. The control group (NOFERT) represents an unfertilized monoculture of IWG. Could the authors explain the rationale for choosing this specific control group and how it relates to the research objectives?
Explanation added into the Methods section
2. The intercrop treatment involves planting IWG and alsike clover in alternating rows. Could the authors provide insight into the reasoning behind this specific planting pattern and how it might affect interactions between the two species?
Explanation added into the Methods section
4. Is there information on the number of replicates for each treatment and how randomization was applied within the randomized complete block design? Understanding replication and randomization is crucial for assessing the robustness of the experimental design.
Explanation added into the Methods section
5. All references should be in [ ] instead of ( ).
corrected
6. There are so many incomplete sentences. Please complete the sentences. such as in section 3.1.1, “The potential cumulative impact of the above, coupled with the.....”
corrected
7. In some places, authors mentioned kg/ha; in others, they mentioned kg ha-1. Please use the same format throughout the MS.
Corrected
8. In Conclusion remove fullstop from 4.1 .Conclusion corrected The conclusion is too long, even longer than the abstract; it should be shorter and concise.
Conclusion has been condensed.
The phrase "Favorable moisture conditions" is somewhat vague. Consider specifying what exactly "favorable moisture conditions" means or entail for better clarity.
clarified
Reviewer 2 Report
1. What are the specific environmental conditions and regions within Western Canada where intermediate wheatgrass, or Kernza, has shown the most promise in terms of grain yield potential?
2. The article briefly mentions the environmental benefits of Kernza. Could you delve deeper into how the deep root system and perennial nature of Kernza contribute to soil health, erosion control, and carbon sequestration?
3. Considering the long-term sustainability and environmental benefits of Kernza, what policies or incentives could be put in place to encourage more widespread adoption of this crop among Western Canadian farmers?
Minor editing of English language required
Author Response
responses to comments and remedial actions taken are in the attached document.

Reviewer 3 Report
Dear Authors
The focus of this research is on nitrogen management in the production system and the effect that intercropping with a perennial legume can have on grain yield and yield components in western Canada. Three treatments consisting of a non-fertilized control, an interseeded crop with Trifolium hybridum and a fertilized treatment (50 kg N ha-1 post grain harvest) evaluated at four locations in Prairie Canada.
The experiment was conducted at four locations in western Canada over four growing seasons, with locations, soil types seeding dates. In 2019, small plot trials were established.
The experiment was arranged in a randomized complete block design (RCBD) and was designed to test the effects of the presence of nitrogen in both synthetic and organic forms on grain yield characteristics of dual-purpose IWG systems.
Remarks
1) Suggests presenting the results in diagrams?
2) The conclusions are coherent and concrete and follow directly from the research carried out. However, they can be supplemented by the most important result of the analysis.
3) The references are appropriate and their number (54 items) optimal. However, they require a thorough editorial revision.
4) Other comments are included in the manuscript

Author Response
Responses to comments and remedial actions taken are in the attached document.

Reviewer 4 Report
First of all, the amount of data presented is not sufficient to be published in the journal agriculture. There are several parameters to be included under the yield potential indices to bring effective relationship and yield potential outcomes. In present form, the article is looking so simple. Additionally, the relationship studies has not been performed and discussed through scientific statistical outputs. Treatment combinations are not so impressive. Year to year variations and discussion in consideration with climatic data has not been well established. There are several drawbacks in the manuscript and quality of presentation (in the form of tables or figures).
Moderate english corrections needed.
Author Response
Responses to the comments and suggestions are in the attached document and in the manuscript itself.
Limits to the number of people able to visit sites during 2020 and 2021, especially government run sites limited measurements and data collection, in part due to restrictive times for entry, exacerbated by the distance some of the sites. These are also the first studies in our region using adapted materials.
Analysis was added to the manuscript and changes were made to the results presentation. (PROC CALIS for modelling of grain yield).
As this is the first regional study on IWG grain production, conditions used in the production mimicked the nursery conditions as we were relatively confident of the ability to be productive under these conditions.

Round 2
Reviewer 4 Report
Authors justified my queries and tried to attempt and address my suggestions. In the line with the adoption of corrections also suggested by the other reviewers, manuscript has come to pretty scientific form. Article may be accepted for publication.
Minor editing may be required.